# Electricity Consumption and Operational Carbon Emissions of European Telecom Network Operators

Dag Lundén [1,*], Jens Malmodin [2,*], Pernilla Bergmark [2] and Nina Lövehagen [2]

1 Telia Company AB, 169 94 Solna, Sweden
2 Ericsson Research, Ericsson AB, 164 83 Stockholm, Sweden; pernilla.bergmark@ericsson.com (P.B.); nina.lovehagen@ericsson.com (N.L.)
* Correspondence: dag.lunden@teliacompany.com (D.L.); jens.malmodin@ericsson.com (J.M.)

**Abstract:** This study presents operational electricity consumption and greenhouse gas emissions for named European telecom network operators during 2015–2018. These results are also compared to data for 2010–2015. The study provides an extensive primary data set, collected from European Telecommunication Network Operators (ETNO) members, covering operations in Europe and beyond, providing data with higher granularity than publicly available sources. The collected data set corresponds to roughly 36 percent of European subscriptions and 8 percent of global subscriptions. This data set was used to calculate the aggregated annual electricity consumption for the assessed operators, as well as associated subscription intensities, in total, for Europe and per network type. Moreover, aggregated electricity-related carbon emissions and emissions from other sources were calculated. Finally, estimates were made for the overall network operation in Europe for 2018 and 2020. The study concludes that the electricity consumption and number of subscriptions for the reporting telecom network operators remained nearly constant (+1 percent and −3 percent, respectively) between 2015 and 2018, while data traffic increased by a factor of three. For the extended period of 2010–2018, the electricity consumption per subscription remained quite stable, slightly below 30 kWh/subscription despite substantial data traffic growth (by a factor of 12).

**Keywords:** ICT sector; mobile; fixed; PSTN; wireless; cellular; operators; connectivity; GHG emissions; carbon dioxide





## 1. Introduction

The access to telecommunication, both fixed and mobile, is a prerequisite for the ongoing digitalization of society. Assessments of the footprint of the Information and Communication Technology (ICT) sector are prevalent and show that it represents a relatively small share of the global electricity consumption and carbon emissions, even when including its full life cycle [1–5]. However, the results of such studies are not directly comparable, as they differ in method and scope. The ICT impact study performed by a consultant commissioned by the European Commission included a projection of the future carbon footprint based on the annual electricity use per part of the ICT sector using data from several sources [5]. OECD/IEA explored digitalization and energy based on previously published data and emphasized the importance of the future use of renewable energy sources and continued energy efficiency improvements [6]. More recently, in 2019, 4E TCP published a high-level model including the global energy consumption of wireless access networks based on secondary data [7]. In a more detailed study, Bieser et al. investigated the mobile communication networks for 2020 and 2030, where the energy consumption of different components was calculated as eighty percent of a nominal energy consumption and the average quantity per site [8]. Few studies have contained primary and measured data. However, in 2014, Van Heddeghem et al. published trends in the worldwide ICT electricity consumption for 2007–2012 using data from a sample of operators worldwide [9].

The KTH Royal Institute of Technology published a similar study by the authors reflecting a 2010–2015 data set that collected data from a number of European-based operators [10].

The ICT sector is dynamic and swiftly developing. For this reason, regular data collection is needed to maintain an up-to-date understanding of its footprint. ICT systems are also complex to model, making it important to collect data from their actual operations. Such data are considered more accurate than any modeled estimates that are based on one or a few parameters [11]. This complexity is, in particular, applicable in telecom networks, as explained and outlined in References [4,12].

The purpose of this study is to provide data regarding ICT operators and networks for actual operational electricity consumption and carbon emissions for the period 2015–2018. As data sets on electricity consumption and operational carbon emissions of existing telecom networks are rarely put forward in the literature, this paper seeks to fill this gap. To the best of the authors' knowledge, no such data set based on primary data collected directly from operators has previously been published for this period. Furthermore, the study investigates the situation in 2020 and explores how telecom networks' energy consumption and operational carbon emissions are impacted by growth in data traffic and subscriptions. The results from this study could be used as a reference in studies investigating the carbon footprint of the entire ICT sector or parts thereof. The use of ICT in all other sectors is paramount and brings opportunities for reducing their environmental impacts. In this respect, telecom networks provide the connectivity that enables data transfer and optimization of systems in real time. However, such effects are outside the scope of this paper.

The authors' previous study observed that, between 2010 and 2015, the electricity consumption grew by 31% for the assessed networks, and the operational carbon emissions grew by 17% [10]. This could be compared to the increase in the number of subscriptions of around a third (from 6.7B to 9.0B). The analysis of data traffic performed for some operators found a strong growth between +235% (3.35×) and +500% (6×). These operators also showed a more moderate increase in electricity consumption compared to the overall data set, with changes between −9% and +11%.

Due to increased openness and transparency among the assessed telecom network operators, and in contrast to the 2010–2015 study [10], this study can name the contributing operators, as well as list the countries of the network operations. Compared to publicly available data from these operators, the primary data set collected for this study provides a higher granularity, which allows for exploring the detailed distribution of operational electricity and carbon emissions between fixed and mobile networks, data center operations, offices, and stores. The contribution of each operator at this granular level cannot be shared, but the granular data set has been validated for the operators' publicly available data, as described in Section 3.5.

The paper is structured as follows: Section 2 introduces the methodology, definitions, and data sources. Section 3 presents the results, including the total and allocated electricity consumption, related carbon emissions, other emissions of the assessed operators, and corresponding intensities for 2015–2018, as well as derived estimates for all Europe and for 2020. This is followed by the discussion in Section 4 and, finally, the conclusion in Section 5.

## 2. Materials and Methods

This study derives the operational electricity consumption (referred to as 'electricity consumption') and the associated greenhouse gas emissions measured in $CO_2$ equivalents (in this paper, referred to as 'carbon emissions', although considering all greenhouse gas emissions) of a subset of European operators, including the operation of networks, operators´ supporting activities, and operation of the operators´ data centers (the latter two referred to as 'overhead'). In line with the principles for company reporting established by the GHG protocol, the electricity supply chain and distribution losses are not included in these carbon emissions but are accounted for separately [13]. Consequently, if these results should be reused for deriving the overall carbon footprints of networks, the energy

supply chain emissions would need to be added to those representing the emission related to the use of this energy.

From the life cycle perspective, this study focuses on the use stage, i.e., the emissions of other life cycle stages of the telecom networks, such as electricity consumption and carbon emissions related to the manufacturing and disposal of hardware, are not included. Based on Reference [4] (Table IV.1), emissions related to other phases of the life cycle roughly represent an additional 16% if a world average energy mix is assumed.

For this study, an extensive primary data set was collected during 2019 through a survey distributed to members of the European Telecommunications Network Operators´ association (ETNO) to collect data for 2015–2018 with a higher granularity than in their public annual reports. The survey questionnaire can be found in the Supplementary material. The year 2015 was chosen as the starting year as the study was intended to complement the corresponding previous study by the authors covering 2010–2015 [10]. The survey included above fifty parameters to be reported for each year of the studied period. The collected data covered the company data and characteristics, a number of subscriptions of different kinds, amount of data traffic (voluntarily allocated to fixed and mobile subscriptions), purchased electricity, own generation of electricity, amount of renewable electricity, details on colocations, carbon emissions (market-based and location-based), electricity consumption detailed per activity and network type, diesel consumption, other energy consumptions, and fuels detailed per activity and the carbon emissions for these. This is further detailed in the supplemental materials. A full year reporting approach was applied in this study. Consequently, recalculations were made for those operators with reporting based on split calendar years.

Figure 1 shows a total research flowchart presenting the different activities and results of the study. After scoping and data collection, including a data quality check leading to an iterative process for collecting data, the overall electricity consumption and the related carbon emissions were summarized for each studied year for all responding ETNO operators (reflected in Section 3.1). Next, the electricity consumption directly used for the operation of the networks was aggregated in the same way. These values were also derived for operations within and beyond Europe. Following this, the electricity consumption per network type and overhead activity for all reporting operators were summarized (as presented in Section 3.2). Reflecting these values and the summarized number of subscriptions, the electricity consumption per subscriber intensities were derived (Section 3.3). Additionally, the carbon emission data collected for other energies and fuels were summarized for all reporting operators per type of activity (Section 3.4).

A validation of the data collected for this study was performed for electricity consumption and the use of renewables by collecting the publicly reported values for the individual operators and comparing them one by one at an aggregated level to this study´s data set to investigate the alignment (Section 3.5). To derive the trends of 2010–2018, the derived electricity and carbon intensities per subscription were compared to the corresponding values of a similar data set representing the development of 2010–2015 (Section 3.6). A similar comparison for data traffic intensities was attempted but could not be implemented due to the lack of an agreed upon definition for measuring data traffic. Instead, both data traffic and electricity consumption were indexed and set in relation to the data for 2015 (Section 3.7). Since the data for 2020 has emerged after the ETNO data set was collected for 2015–2018, publicly available data for 2020 were collected and analyzed with regards to similarities and differences compared to the 2018 data (Section 3.8).

Finally, the ETNO data set was used as a basis for estimating the overall electricity consumption and the associated carbon emissions for operators in Europe, specifically for EU-27, UK, Switzerland, and Norway based on extrapolation reflecting the number of subscriptions for 2018 (Section 3.9). To develop a similar estimate for 2020, an additional but more limited data set representing additional operators was collected. The combined 2020 data set was then extrapolated based on the number of subscriptions to estimate

EU-27, UK, Switzerland, and Norway (Section 3.10). Table 1 details the network types and activities covered by this study. These are also shown in Figure 2.

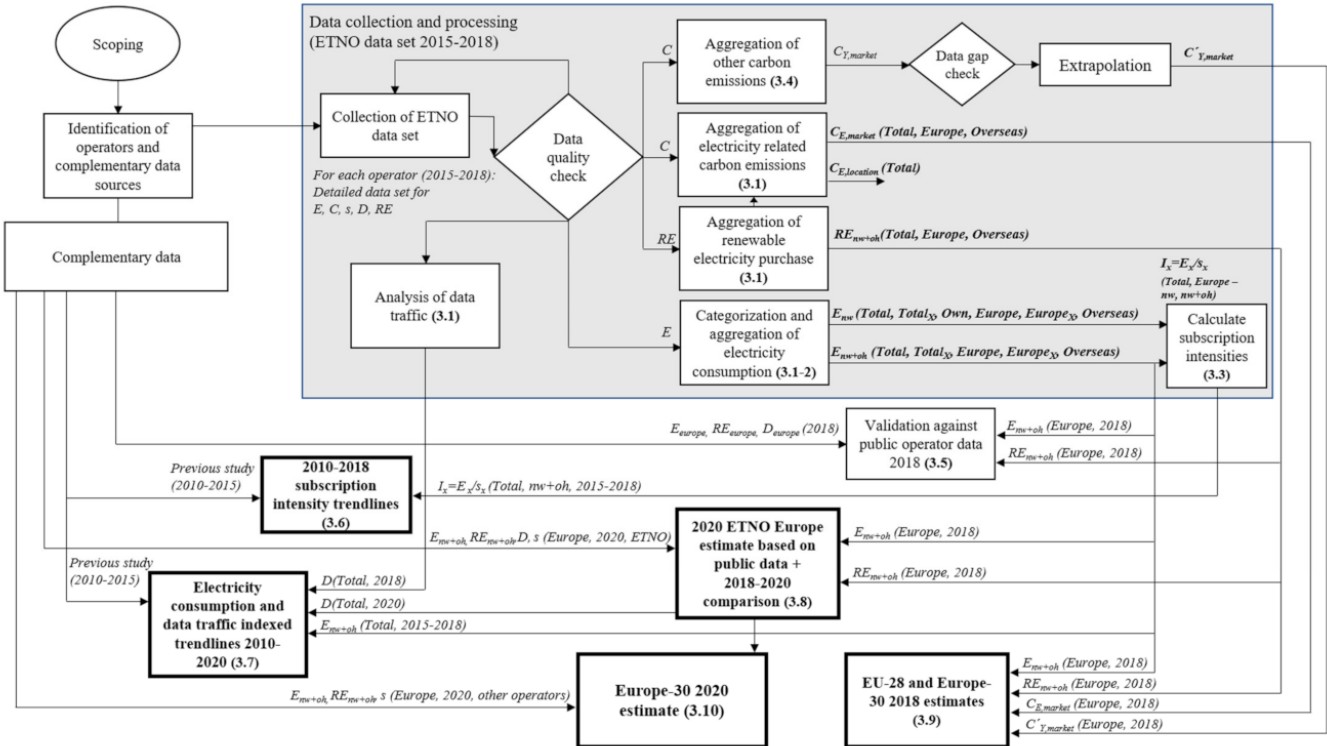

**Figure 1.** Research flowchart outlining how the primary ETNO data set is combined with other sources to derive the results of the study. Text in bold denotes results. Note that some results are used as input for later steps. Sections in the article are indicated within parentheses.

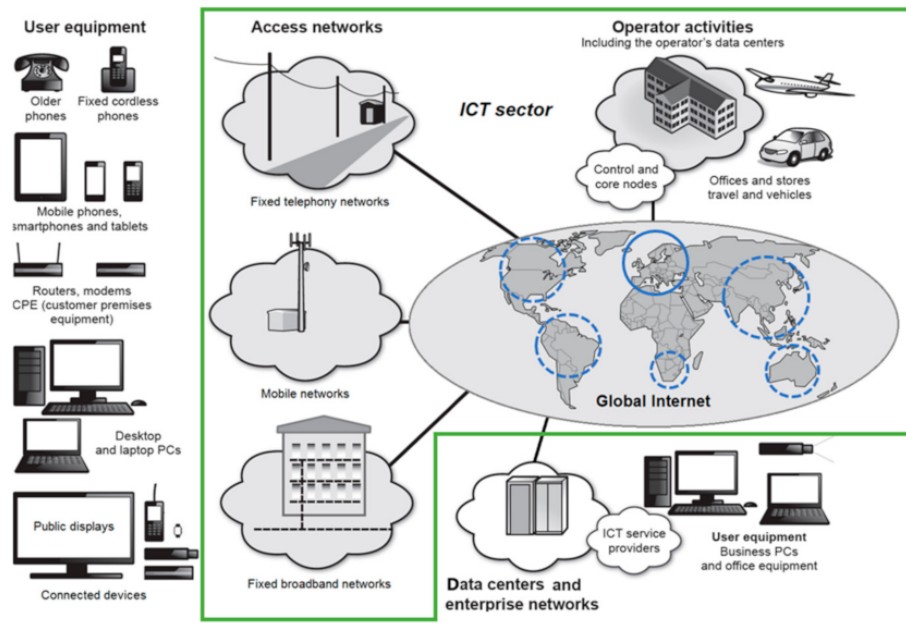

**Figure 2.** The scope of this report (inside the green outline) compared to the scope of the full ICT sector. Geographically, a major part of the data set represents Europe (the blue circle), but operations outside of Europe are covered by some of the reporting operators (dashed blue circles).

**Table 1.** Included and excluded network components and activities.

| Included Network Types: |
| --- |
| Fixed telephony access networks (aka PSTN [1]/POTS [1]) <br> Fixed broadband access networks <br> Mobile access networks (incl. all generations and services such as voice, data, and M2M [1]) <br> Data transmission and IP-core network [1] <br> Telecom network operators' data centers <br> Cable TV networks and associated provision of broadband if operated by network operators [2] <br> Additional services provided by network operators (such as IPTV [1,3] and VoIP [1]) |
| **Included supporting activities:** |
| Operation of offices and stores <br> Business travel and field service operations <br> Local production of electricity (including diesel generators and solar cells both for off grid and power backup solutions) |
| **Not included:** |
| Enterprise networks (aka intranet) provided to customers <br> Home networks / CPE [1] (e.g., modems, gateways, etc.) |

[1] PSTN: Public Switched Telephone Network, POTS: Plain Old Telephone Service, M2M: Machine to Machine, IP: Internet Protocol, IPTV: Internet Protocol Television, VoIP: Voice Over IP, and CPE: Customer Premises Equipment. [2] Cable TV (CATV) networks are outside the scope of this study. However, for some network operators that operate local CATV networks, their reported electricity consumption includes these networks. Fixed broadband over CATV is estimated to consume about the same amount of electricity per subscription as fixed broadband over a digital subscriber line (DSL) or fiber [10]. However, their impact and total share is small in comparison to the total network operations. [3] Some network operators provide their own IPTV services, which are then also included in the telecom network operators' electricity consumption. Depending on the operators' reporting structure, this electricity consumption is either allocated and reported as a part of the data transmission and IP core network and/or as part of telecom data centers' electricity consumption.

The contributing ETNO members include 15 named telecom network operators with headquarters in Europe, representing operations in 21 European countries. In addition, their operations in 27 countries overseas were included. The full list of countries is outlined in Table 2. However, as two of the operators reported their overall operations in an aggregated format without country details, the actual number of countries might be higher than specified. The total number of telecom networks covered by the data set amount to 59, where several operators may run parallel networks in the same country.

**Table 2.** Countries where participating ETNO Operators operate.

| Continent | Countries | | | |
| --- | --- | --- | --- | --- |
| Europe | Belgium <br> France <br> Ireland <br> Netherlands <br> Slovenia <br> UK | Denmark <br> Germany <br> Italy <br> North Macedonia <br> Spain | Estonia <br> Greece <br> Lithuania <br> Norway <br> Sweden | Finland <br> Hungary <br> Luxembourg <br> Portugal <br> Switzerland |
| America | Argentina <br> Ecuador <br> Uruguay | Brazil <br> Indonesia <br> USA | Chile <br> Mexico <br> Venezuela | Colombia <br> Peru |
| Africa | South Africa | | | |
| Asia | Bangladesh <br> India <br> Pakistan <br> Turkey | China <br> Japan <br> Philippines <br> Vietnam | Dubai <br> Malaysia <br> Singapore | Hong Kong <br> Myanmar <br> Thailand |
| Oceania | Australia | | | |

The following network operators have contributed to the study (city of head office within parentheses): Altice (Lisbon), BT (London), Cosmote (Athens), DNA (Helsinki), DT

Germany (Bonn), DT Maygar (Budapest), Elisa (Helsinki), KPN (Rotterdam), Proximus (Brussels), Swisscom (Bern), Telecom Slovenia (Ljubljana), Telefonica (Madrid), Telenor (Oslo), Telia Company (Stockholm), and Telecom Italy (Rome).

In this study, the aggregated data set collected from the listed ETNO operators for their operations in Europe is referred to as 'ETNO Europe' data. Some of the operators have also reported data from operations outside of Europe, and these data are referred to as 'ETNO overseas' data. When referring to both these categories, we use the term 'ETNO Total' data. Moreover, the reporting operators are referred to as the 'ETNO Operators'. Note that this refers to operators active in ETNO´s Sustainability working group, plus additional large ETNO operators, which is a subset of ETNO´s operator membership. The subscriptions of the ETNO data set compared to the global and European overall subscriptions are shown in Tables 3 and 4, respectively.

**Table 3.** ETNO Total subscription data for 2018 (and 2015) compared with the global subscriptions.

| Subscriptions | ETNO Total [1] 2018 (2015) (Million) | Total Global 2018 [14] [2] (Million) | ETNO´s Share of Global Subscriptions 2018 |
|---|---|---|---|
| PSTN subscriptions<br>VoIP subscriptions | 82 (95)<br>25 (16) | 942 | 8.7% [3] |
| Mobile subscriptions [4] | 664 (694) | 7972 | 8.4% |
| Fixed broadband subscriptions | 72 (64) | 1065 | 6.7% |
| Total subscriptions | 843 (869) | 9979 | 8.2% [3] |

[1] For the countries covered, see Table 2. [2] ITU statistics for mobile broadband exist as well, but this category is not used in this paper due to a lack of consistent definition and reporting. [3] The share of global subscriptions conservatively excl. VoIP, although incl. in the reference value [14]. [4] M2M subscriptions have been excluded as far as the data set allows.

**Table 4.** ETNO Europe subscription data for 2018 (and 2015) compared with the European subscriptions.

| Subscriptions | ETNO Europe [1] 2018 (2015) (Million) | EU-28 [2] 2018 [15–17] (Million) | Europe-30 [3] 2018 [15–17] (Million) | ETNO Europe's Share of Europe-30 2018 |
|---|---|---|---|---|
| PSTN subscriptions<br>VoIP subscriptions | 55 (68)<br>25 (16) | 202 | 206 | 27% [4] |
| Mobile subscriptions [5] | 255 (265) | 620 | 637 | 40% |
| Fixed broadband subscriptions | 59 (53) | 181 | 187 | 32% |
| Total subscriptions | 394 (402) | 1003 | 1030 | 36% [4] |

[1] For the countries covered, see Table 2. [2] UK was still a member of the EU during 2018. [3] Covers EU-27 and UK, Norway, and Switzerland. [4] The share of subscriptions conservatively excl. VoIP, although incl. in the reference values [15–17]. [5] M2M subscriptions have been excluded as far as the data set allows.

For 2018, the aggregated data set of the reporting operators covered about 843 million subscriptions, corresponding to about 8.2% of the global total estimate in 2018 of 9979 million, including fixed telephony (PSTN—public switched telephone network and VoIP—voice over Internet Protocol), mobile, and fixed broadband subscriptions. The telecom network usage by Machine to Machine (M2M) and IPTV (television over IP) subscriptions is covered within the electricity consumption and related carbon emissions. However, these subscriptions are not relevant when deriving intensity metrics, as these have a limited impact on the electricity consumption, since they are less significant for dimensioning the network. M2M over the mobile network can, from a subscription quantity perspective, be either included or not, depending on the telecom operators' subscription offerings. Moreover, the networks operators' cable television (CATV) electricity consumption, if existing, is covered

in the reported energy numbers. However, CATV subscription numbers are not considered when calculating shares of the overall subscriptions.

## 3. Results

In this section, aggregated numbers for the reporting ETNO operators are presented both as absolute values and intensities, and the overall data set is validated against the operators´ publicly available numbers. The data set is also compared to 2010–2015 data, as well as to 2020 data. Finally, estimates for the entire Europe-30 region are presented.

### 3.1. Electricity Consumption and Related Carbon Emissions of ETNO Operators

The ETNO Operators' aggregated electricity consumption, including both electricity consumed by networks and other electricity consumption, originates to a large extent from the purchase of grid electricity. Only a limited share originates from the operators´ own onsite-generated electricity production. During the assessed period, their total electricity consumption, including all sources, increased slightly from 20.8 TWh in 2015 to 21.0 TWh in 2018. For 2018, 14.5 TWh of this consumption referred to ETNO Europe, which was quite similar to the reported 14.3 TWh of the electricity consumption for 2015. For the operation of telecom networks, the electricity consumption increased from 17.7 TWh to 18.4 TWh between 2015 and 2018. Thus, the network represented 85% and 87% of the overall purchase of electricity by the network operators, respectively. In addition, the network operators' own electricity generation onsite increased but from very low levels. In 2018, this electricity corresponded to about 250 GWh or 1.2% of the consumed 21.0 TWh, up from 150 GWh in 2015. During the period 2015–2018, the share of purchased renewable electricity increased significantly and represented 55% of the total grid electricity consumption in 2018 (up from 45% in 2015), as depicted in Figure 3. Depending on the electricity mix and the availability of renewable electricity on the market, the resulting climate impact varies substantially between countries and develops over time.

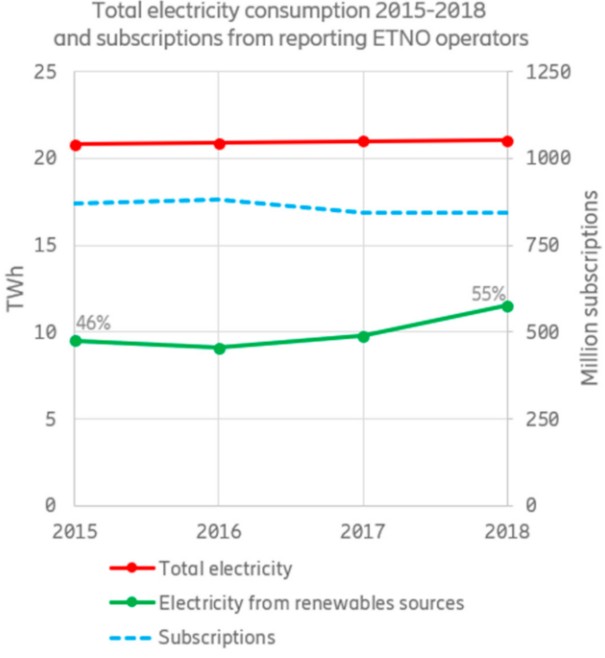

**Figure 3.** Total electricity consumption reported by ETNO Operators for 2015–2018, including the share of consumed renewable electricity.

In this paper, 'renewable electricity' refers to guarantees of the origin available at the local markets of the operators; thus, it includes all kinds of renewable electricity purchase agreements between operators and utility providers. The specific electricity standards and definition of 'renewable' vary depending on the country and region. Moreover, the ETNO

Total data set covers operations both within Europe and regions where renewable electricity is scarcer, so the share of renewables varies greatly with the geographical scope. The overall development for 2015–2018 of the assessed ETNO Operators´ electricity consumption reported the use of renewables, and the number of subscriptions is summarized in Figure 3.

Table 5 outlines the reporting ETNO operators´ carbon emission details for 2018, including both the actual purchases of renewables (aka market-based method) and the emissions corresponding to the combination of electricity sources (electricity mix) in the local electricity grid (location-based method) [18]. These values exclude emissions in the electricity supply chain and losses in the electricity distribution, which are accounted for separately (see Section 3.4).

**Table 5.** Total carbon emissions from the electricity consumption of 2018 based on (A) the reported actual conditions and (B) average country conditions.

| **Reported Electricity Consumption:** | |
| --- | --- |
| Total electricity reported | 21.0 TWh |
| whereof self-generated electricity (renewable and nonrenewable) | 0.25 TWh |
| whereof purchased renewable grid electricity | 11.3 TWh |
| A. Reported carbon emissions based on actual use of renewables (market-based method): | |
| Average renewable electricity emission factor (EF) | 0.01 kg $CO_2$/kWh |
| Average non-renewable electricity EF | 0.37 kg $CO_2$/kWh |
| Reported carbon emissions due to total electricity consumption | 3626 kton $CO_2$ |
| whereof due to reported renewable electricity | ~100 kton $CO_2$ |
| B. Carbon emissions using country-based emission factors (location-based method): | |
| Emissions without consideration of the purchase of renewables based average country EFs [1] | 6700 kton $CO_2$ |

[1] Country EFs are based on energy mixes that include both renewables and nonrenewables.

Figure 4 outlines the aggregated carbon emissions following both the market-based and location-based methods for the full period of 2015–2018. There are substantial regional differences in the usage of renewables, and the share of renewable electricity supplies is higher in Europe compared to operations in the rest of the world for ETNO Operators, as shown in Figure 5.

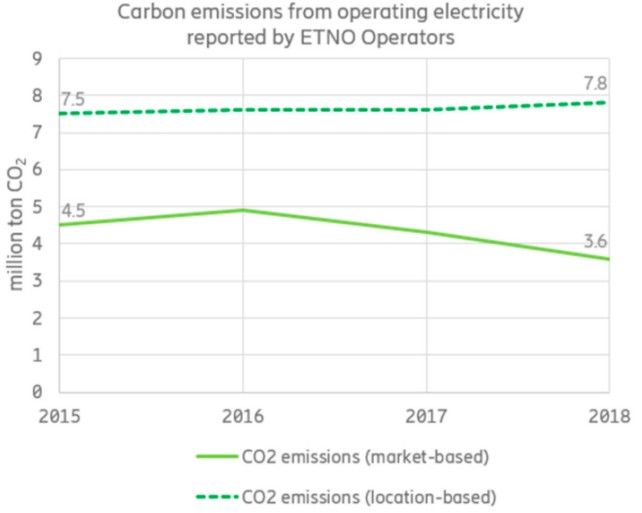

**Figure 4.** Carbon emissions due to the total operational electricity used by the ETNO Operators for 2015–2018 as reported, including the purchased and own production of electricity. The dotted line shows emissions that would have occurred without the purchase of renewables, i.e., following the country grid averages. Electricity supply chain and losses are not included.

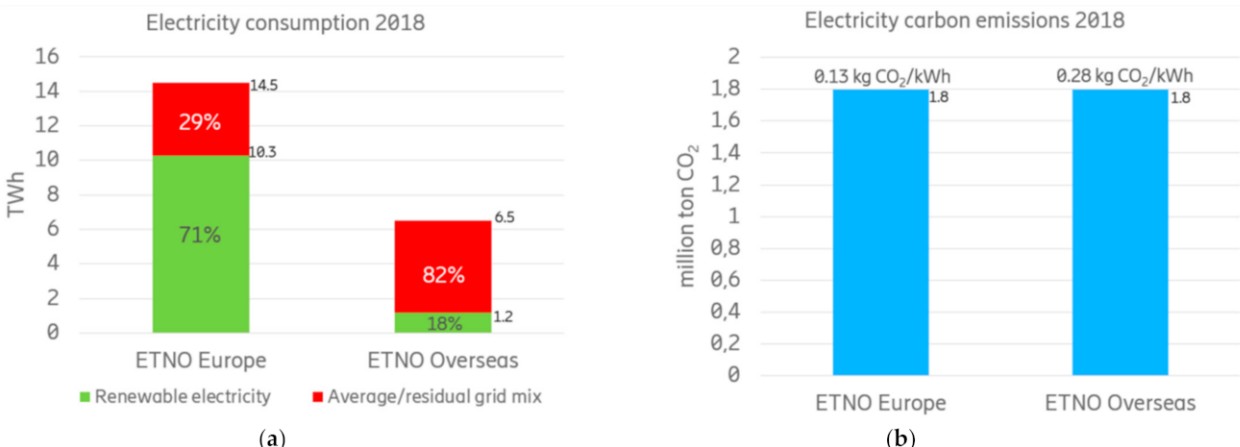

(a)　(b)

**Figure 5.** (**a**) The reporting network operators' share of renewable electricity in 2018 for ETNO Europe data and for the ETNO Overseas data operations (as reported by the network operators; see Table 2). (**b**) Corresponding electricity carbon emissions with the resulting emission intensity stated.

As depicted in Figure 5, the share of renewable electricity among the reporting ETNO Operators equals about 71% within Europe, whereas the overseas operations have a share of 18%. The reason for this difference is their more frequent purchasing of renewable electricity in Europe due to a more mature electricity market where renewable electricity is less expensive and more widely available.

### 3.2. Electricity Consumption Per Network Type

In Figure 6, the trends for aggregated electricity consumption are presented per network type, as well as for network operator data center operations, stores, and offices, for the period 2015–2018. From an electricity consumption perspective, PSTN, as well as offices, stores, and others, declined during the period, while mobile and fixed broadband networks increased. This reflects a shift away from copper-based fixed access and PSTN services. Detailed aggregated electricity consumption numbers and consumption details per network type for 2015–2018 can be found in Table 6.

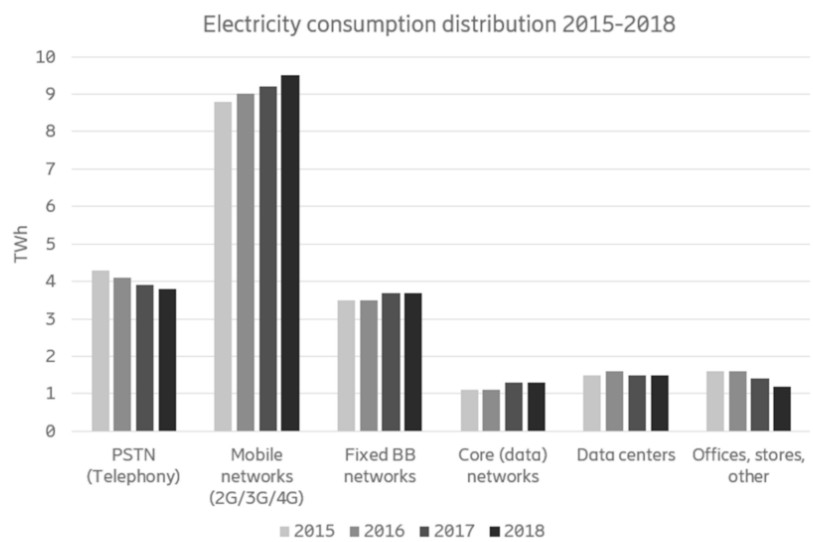

**Figure 6.** ETNO Operators' total aggregated electricity consumption and trends per network type and overhead activity for the reporting ETNO operators of 2015–2018. These numbers exclude the electricity supply chain and distribution losses.

**Table 6.** Detailed aggregated electricity consumption and trends per network type and overhead activity for the ETNO Operators.

| | Electricity Consumption (TWh) | | | |
|---|---|---|---|---|
| | **2015** | **2016** | **2017** | **2018** |
| PSTN access network | 4.3 | 4.1 | 3.9 | 3.8 |
| Mobile access networks | 8.8 | 9.0 | 9.2 | 9.5 |
| Fixed broadband networks | 3.5 | 3.5 | 3.7 | 3.7 |
| Data and core networks | 1.1 | 1.1 | 1.3 | 1.3 |
| Operators' data centers | 1.5 | 1.6 | 1.5 | 1.5 |
| Offices, shops, stores etc. | 1.6 | 1.6 | 1.4 | 1.2 |
| Total | 20.8 | 20.9 | 21.0 | 21.0 |

### 3.3. Electricity Consumption Per Subscription

To set the above trends in relation to the number of subscriptions—a proxy for the number of users—the intensities per subscription are derived. Figure 7 shows the ETNO Operators' total average electricity consumption per subscription, as well as the intensities for mobile and fixed networks for the ETNO Total and ETNO Europe data sets. The values include the electricity consumption for the operators' overhead operations, which has been allocated based on subscription volumes as follows:

(i)     Energy consumption related to offices, stores, etc.: 60% mobile, 30% fixed broadband, and 10% PSTN.

(ii)    Data network operations: 55% fixed broadband, 35% mobile, and 10% PSTN. Half of the overhead electricity consumption is related to onsite installations and is divided equally between the three network types, and the remaining half is related to the data traffic, which is shared between mobile and fixed broadbands.

(iii)   Operators' data center operations: 55% mobile and 45% fixed broadband. The allocation is based on the conditions of the data set. However, there are great variations between the operators. As in Sections 3.1 and 3.2, the electricity supply chain and distribution losses are not included.

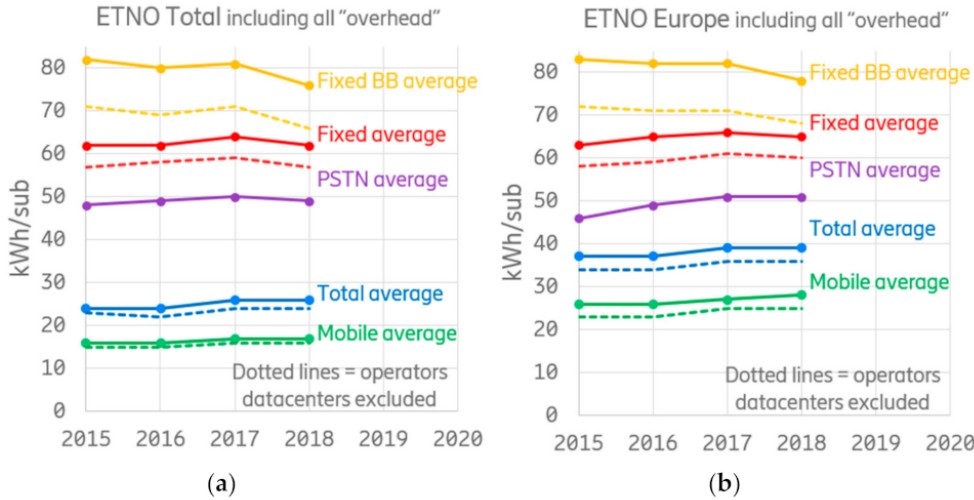

**Figure 7.** Electricity consumption per subscription for fixed, mobile, and combined subscriptions of the reporting ETNO Operators for (**a**) their overall reported operations and (**b**) those within Europe. The solid lines include all overhead, whereas the dotted lines exclude the operators' data centers. Electricity supply chain and losses are not included.

Overall, there are only limited changes in the energy intensity per subscription between 2015 and 2018, as shown in Figure 7. Additionally, both for ETNO Total and ETNO Europe data sets, the energy intensity is higher for fixed communication than for mobile. Moreover, for the assessed ETNO Operators, the electricity consumption per mobile subscription for Europe is larger than for their total operations. The fixed averages are similar within Europe and for the overall data set.

Next, Figure 8 shows the electricity per subscription intensities for networks only (excluding all overhead). Since the allocation of the overhead between different types of networks in Figure 7 is model-based, the values for specific network types in Figure 8 are considered to have a higher accuracy. For the total averages, there is no difference in accuracy between Figures 6 and 7.

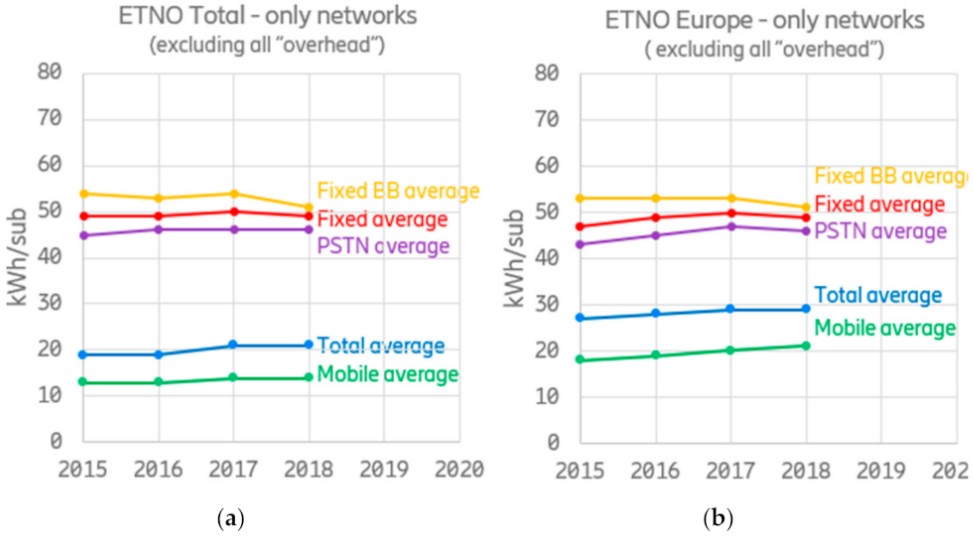

**Figure 8.** Electricity consumption for the network per subscription for fixed broadband, PSTN, and averages for the fixed, mobile, and total based on the (**a**) ETNO Total and (**b**) ETNO Europe data from all reporting network operators during 2015–2018.

The electricity per subscription and network type, both including and excluding overhead, are detailed in Tables 7 and 8 for the ETNO Total and ETNO Europe, respectively, for the period 2015–2018.

**Table 7.** Electricity consumption per subscription and network type for the ETNO Total data set, including and excluding overhead.

| | 2015 (kWh/sub) | 2016 (kWh/sub) | 2017 (kWh/sub) | 2018 (kWh/sub) |
|---|---|---|---|---|
| With Overhead Including Operators' Data Centers | | | | |
| PSTN average [1] | 48 | 49 | 50 | 49 |
| Fixed BB average | 82 | 80 | 81 | 76 |
| Fixed average | 62 | 62 | 63 | 62 |
| Mobile average | 16 | 16 | 17 | 17 |
| Total average | 24 | 24 | 26 | 26 |
| Without overhead (networks only) | | | | |
| PSTN average [2] | 45 | 46 | 46 | 46 |
| Fixed BB average | 54 | 53 | 54 | 51 |
| Fixed average | 49 | 49 | 50 | 49 |
| Mobile average | 13 | 13 | 14 | 14 |
| Total average | 19 | 19 | 21 | 21 |

[1] If this would also consider VoIP subscriptions, the values for 2015–2018 would be 41, 40, 39, and 38 kWh/subs. [2] If this would also consider VoIP subscriptions, the values for 2015–2018 would be 39, 38, 37, and 36 kWh/subs.

**Table 8.** Electricity consumption per subscription and network type for the ETNO Europe data set.

|  | 2015 (kWh/sub) | 2016 (kWh/sub) | 2017 (kWh/sub) | 2018 (kWh/sub) |
|---|---|---|---|---|
| **With Overhead Including Operators' Data Centers** | | | | |
| PSTN average [1] | 46 | 49 | 51 | 51 |
| Fixed BB average | 83 | 82 | 82 | 78 |
| Fixed average | 63 | 65 | 66 | 65 |
| Mobile average | 26 | 26 | 27 | 28 |
| Total average | 37 | 37 | 39 | 39 |
| **Without overhead (networks only)** | | | | |
| PSTN average [2] | 43 | 45 | 47 | 46 |
| Fixed BB average | 53 | 53 | 53 | 51 |
| Fixed average | 47 | 49 | 50 | 49 |
| Mobile average | 18 | 19 | 20 | 21 |
| Total average | 27 | 28 | 29 | 29 |

[1] If this would also consider VoIP subscriptions, the values for 2015–2018 would be 38, 38, 37, and 35 kWh/subs. [2] If this would also consider VoIP subscriptions, the values for 2015–2018 would be 35, 35, 34, and 32 kWh/subs.

### 3.4. Operational Carbon Emissions from Other Sources Than Electricity Consumption

Sections 3.1–3.3 focused on the electricity consumption and the associated carbon emissions. However, electricity is not the only source of operational carbon emissions for operators. Backup fuel, travel for network management, field maintenance activities, etc. also contribute. Figure 9 shows such emissions, as well as the fuel and energy supply chains (including the electricity supply chain) for the assessed ETNO Operators, including both the ETNO Europe and ETNO Overseas data sets. Note that these data do not represent the full-value chain perspective of the operators, as only emissions associated with the direct operations are explored in this study. Although the emissions associated with the use of electricity (as presented in Section 3.1) represent the largest contribution, the aggregated effects of the other emissions cannot be neglected, especially not in a scenario with a substantial active purchase of renewable electricity supply.

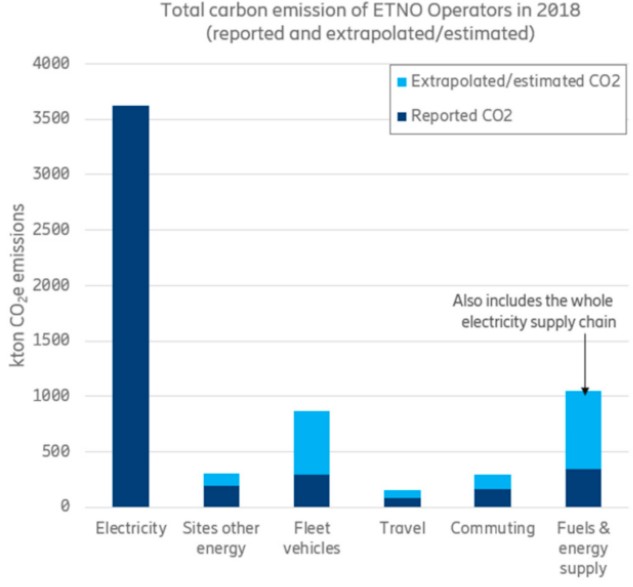

**Figure 9.** Carbon emissions related to the ETNO Operators' usage of electricity in 2018 (left bar) and operational carbon emissions from other sources. Note that, according to the report by the ETNO Operators, emissions related to the supply chain of renewable electricity are included in the electricity bar instead of the fuels and energy supply bar.

The reporting quality of nonelectric energy at sites, fuel usage of vehicle fleets, business travel, and commuting vary between the assessed ETNO Operators. Thus, Figure 9 was derived by extrapolating the reports of a limited subset of the operators (roughly one-third) and, mainly, their European operations to estimate the overall contributions from the ETNO Operators. The extrapolations were based on the number of employees (for travel and commuting), subscriptions (for fleet vehicles), and individual estimates (for nonelectric energy usage at sites). The corresponding carbon emissions in total and per subscription for the year 2018 are detailed in Table 9.

**Table 9.** Carbon emissions for the year 2018 related to the ETNO Operators' usage of electricity (total) from other sources, as well as carbon emission intensities per subscriptions.

| | Reported Carbon Emissions (kton) | Estimated ETNO Total (Extrapolated) (kton) | Carbon Emission Per Subscription (kg/sub) |
|---|---|---|---|
| Electricity (market-based) [1] | 3626 | 3626 | 4.4 |
| Sites other energy | 196 | 300 | 0.24 |
| Fleet vehicles | 294 | 870 | 0.36 |
| Travel | 86 | 150 | 0.11 |
| Commuting | 165 | 290 | 0.20 |
| Fuels and energy supply | 341 | 1050 | 1.3 |
| Total | 4708 | 6286 | 6.6 |

[1] According to the report by the ETNO Operators, emissions related to the supply chain of renewable electricity are counted as electricity instead of fuels and energy supply.

### 3.5. Validation of ETNO Data Set against the Operators' Publicly Reported Values

The ETNO Europe data set used in this study has been compared to the reported ETNO Operators' 2018 Europe values in their public (but less granular) environmental data reports. The publicly reported values are listed in Table 10. The comparison was limited to Europe, as it was hard to find comparable public data for ETNO's overseas operations.

**Table 10.** Reported the 2018 data (as publicly reported by the ETNO operators [1]).

| Operator | Electricity (GWh) | Renewable (GWh) | Data Traffic (EB) | Comments |
|---|---|---|---|---|
| Altice | 322 [19] | 135 [19] | 4.33 [20] | Reported 42% renewables |
| BT | 2757 [21] | 2381 [22] | ~20 [23] | Reported 2019 values due to split financial year |
| Cosmote | 490 [24] | 480 [25] | 0.1 [24] | Greece only |
| DNA | 164 [26] | unclear | | 100% renewable stated but unclear numbers, thus not considered |
| DT—Germany | 2769 [27] | 1702 [27] | ~30 [27] | |
| DT—Maygar | | | | No public data available |
| Elisa | 293 [28] | 271 [28] | 1.54 [28] | |
| KPN | 631 [29] | 631 [29] | 1500% [29] | Excl. sold business |
| Proximus | 368 [30] | 368 [30] | | |
| Swisscom | 485 [31] | 485 [31] | | |
| Telecom Italia | 1859 [32] | 55 [32] | 13.1 [32] | Italy only |
| Telecom Slovenia | | | | No public data available |

**Table 10.** *Cont.*

| Operator | Electricity (GWh) | Renewable (GWh) | Data Traffic (EB) | Comments |
|---|---|---|---|---|
| Telefonica | 2934 [33] | 2934 [33] | 47.3 [34] | Reported 100% renewables for Europe. Data traffic also outside Europe |
| Telenor | 512 [35] | no data | | Total energy reported |
| Telia | 1066 [36] | 1066 [36] | | |
| Total | 14,650 | 10,509 | | |
| ETNO Europe | 14.5 TWh | 10.3 TWh | | The corresponding data from this study included as a reference. |

[1] Note that the 2018 values are taken from the latest possible publicly available reports. The reason for this is that companies may update data for previous years after a final analysis. In addition, when network operators divest parts of their operations or merger with other companies, historical sustainability data have also often been adjusted to align with the new situation [13].

For the total electricity consumption, the ETNO Europe data set of this study is 1% lower than aggregation of the public reporting and, for renewables of ETNO Europe, 2% lower than the public reporting. This comparison includes some differences in scope detailed in the table. For individual operators, the values of this study are in the range of −4% to +1% (except one outlier, which differed +14%) compared to the public reports. As the average differences are limited, the data used in this study are considered well in line with what is reported publicly. Moreover, use of this data set follows a conservative principle, as our values are higher on average. Consequently, although the granular data for each individual ETNO operator used in this study cannot be presented transparently, this section shows that the overall electricity consumption and emission levels are consistent with the publicly available data.

### 3.6. The Extended Trend Line—2010–2018 Development

To investigate the development of operational electricity consumption and the related carbon emissions over a longer time interval (2010–2018), the data set collected for this study was compared to data from a previous study by the authors with 2010–2015 data from operators with headquarters in Europe and some international ones [10]. The resulting graphs are outlined in Figure 10 (electricity per subscription) and Figure 11 (carbon emissions per subscription). The trends are derived per subscription and not for absolute values, since the present study covers a larger set of contributing operators than the previous one and, by that, a larger volume of subscriptions, networks, operator data centers, and offices, resulting in higher and not comparable absolute electricity consumption.

The carbon emissions associated with the average electricity consumption per subscription for the reporting ETNO Operators are shown in Figure 11. The gap in the curve is due to the difference in contributing operators for the two time intervals. The graph shows that, for the assessed operators, the total emissions per subscription decrease. For the period of 2010–2018, the active purchase of renewable electricity increased significantly. The share of renewables has varied over the years. In 2010, a 46% share was reported, but it decreased in 2013 to 31% due to one large operator who temporarily decided not to purchase renewables. However, in 2015, the share was 46% again, and in 2018, it was 55%, as seen in Figure 3.

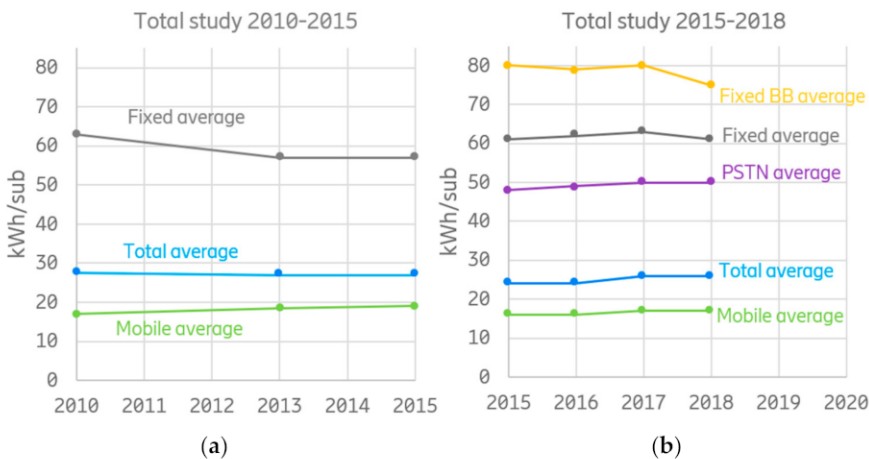

(**a**) (**b**)

**Figure 10.** Electricity consumption per subscription (including all overhead) for all reporting network operators during (**a**) 2010–2015 based on Reference [10] and (**b**) 2015–2018. Note that (**a**) includes diesel for the generation of electricity consumption on site, which is not included in (**b**), corresponding to around 0.1 kWh/sub.

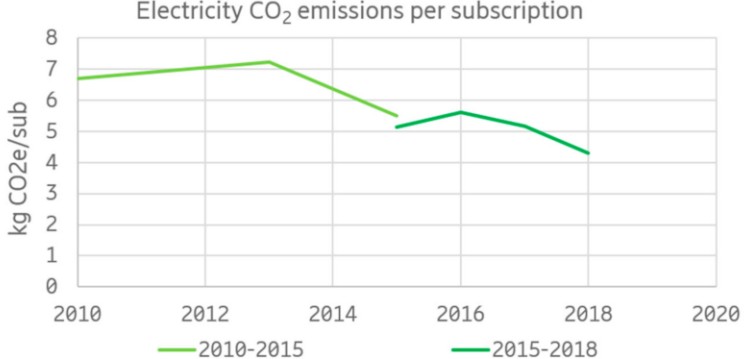

**Figure 11.** Total operational carbon emissions related to electricity per subscription for ETNO operators for 2010–2018, complemented with data for 2010–2015 [10]. Electricity supply chain and losses are not included.

### 3.7. Electricity Consumption in Relation to Data Traffic of 2010–2018

Many of the contributing operators shared details about data traffic when reporting. However, the data traffic in a network can be measured in different ways and at different measuring points, as no clear definition exists on how and where to measure data traffic. Consequently, practices vary between operators, and data traffic could not be aggregated between them. Instead, all data points were first normalized based on the 2015 levels and then aggregated to create an indexed trend curve and scaled so that the 2015 level equals 1. This approach is based on the assumption that individual operator's data traffic measurement practices remain consistent over time. Based on this, the traffic development of each operator was derived and indexed and subsequently aggregated with other operators' indexed data traffic trends.

Figure 12 shows the reporting network operator's electricity consumption (indexed) compared to the indexed data traffic for the ETNO Operators of 2015–2018 (ETNO Total data set). In addition, the reported data set for the period 2010–2015 [10] has been included to derive the trend over a longer time interval. Data traffic was reported to a more limited extent for the period 2010–2015 and only for a few years and by a limited number of operators; thus, only data points for 2010 and 2015 have been included in the figure. The electricity consumption shown in Figure 12 represents the overall electricity consumption for the reporting ETNO operators, hence including the operators' overhead, network, and data center operations (details for each part were outlined in Figure 6).

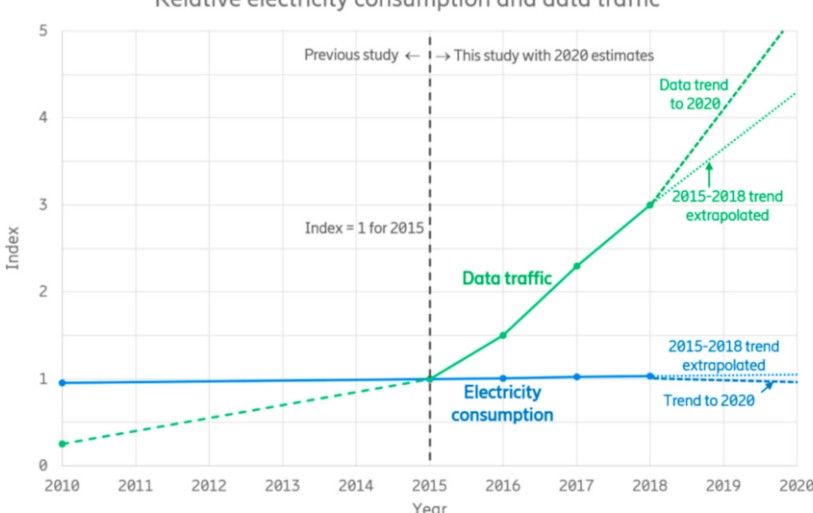

**Figure 12.** Electricity consumption and data traffic for the reporting ETNO operators of 2015–2018 (ETNO Total data set), also including less granular data reported for a more limited number of operators for the period of 2010–2015, as reported in Reference [10]. The electricity consumption and the data traffic for the full period have been indexed in relation to the 2015 level.

As illustrated in Figure 12, there is a substantial growth in processed and transported data and, hence, increased capacity over time. However, this has not been followed by a corresponding increase in electricity consumption. There were two substantial outliers in the data set related to data traffic, but most operators had a reported increase between two and four times between 2015 and 2018. The average is estimated to be around three times, which is slightly higher than expected.

The data set for 2010–2015 showed an electricity growth of 5% during the period. Together with the 2015–2018 increase of 3%, the increase between 2010–2018 amounted to 9% (rounded values). In comparison, the data traffic grew by +300% (four times) in 2010–2015 and by +200% (three times) in 2015–2018, leading to +1100% (12×) between 2010 and 2018. The figure also includes the development of 2018–2020 based on the data for electricity consumption and data traffic described in Sections 3.8 and 3.10. During this period, which also included the initial phase of the COVID-19 pandemic, data traffic grew by an additional +95%, while the electricity consumption decreased slightly.

### 3.8. Electricity Consumption and Operational Carbon Emissions for ETNO Europe 2020

The electricity consumption of the ETNO Operators in 2020 has been estimated for Europe based on the publicly reported data for 2020, including the number of subscriptions. The ETNO Operators' public reports for 2020 are shown and summarized in Table 11 for Europe.

The large data traffic increase experienced in 2020 is assumed due to COVID-19 and the societal lockdowns, but this increase in data seems not to have been reflected in the electricity consumption or number of subscriptions. This is further discussed in Section 4.

A comparison between the publicly reported 2020 data and the aggregated ETNO Europe values for 2018 shows the following:

- Total subscriptions increased 9% (369 million in 2018 to 401 million subscriptions in 2020)
- Electricity consumption decreased 1% (from 14.5 TWh in 2018 to 14.3 TWh in 2020)
- Renewable electricity increased 18% (from 10.3 TWh in 2018 to 12.2 TWh in 2020)
- Share of renewable electricity increased (from 71% in 2018 to 85% in 2020)

**Table 11.** Publicly reported values for Europe from the ETNO Operators for 2020.

| Operator | Electricity (GWh) | Renewables (GWh) | Subscriptions (million) | Data Traffic (EB) | Comments |
|----------|-------------------|------------------|-------------------------|-------------------|----------|
| Altice | 301 [37] | 286 [37] | 7.6 [38] | 7.2 [20] | |
| BT | 2577 [21] | 2577 [21] | 30 [39] | 50 [23] | Only mobile subscriptions |
| Cosmote | 438 [40] | 438 [24] | 12.7 [41] | 5.26 [24] | Only Greece |
| DNA | 153 [35] | no data | 8.8 [42] | | Since 2019, part of Telenor, total energy data |
| DT—Germany | 2718 [27] | 2719 [27] | 62 [43] | 55.8 [27] | No PSTN subscriptions included |
| DT—Maygar | | | | | No public data available |
| Elisa | 294 [28] | 280 [28] | 7.1 [28] | 2.46 [28] | Only mobile data traffic |
| KPN | 573 [29] | 573 [29] | 8.3 [29] | 2100% [29] | |
| Proximus | 352 [30] | 352 [30] | 5.5 [30] | | No PSTN subscriptions included |
| Swisscom | 479 [31] | 479 [31] | 10.3 [44] | | Subscription figures include only Switzerland |
| Telecom Italia | 1561 [32] | 414 [32] | 47 [45] | 20.6 [32] | Italy only, subscriptions as per Sep 2021. |
| Telecom Slovenia | 90 [42] | | 2.9 [42] | | Incl. five small countries outside Europe-30 |
| Telefonica | 2947 [46] | 2947 [46] | 99 [46] | 87.8 [46] | Data traffic includes traffic outside Europe |
| Telenor | 639 [35] | no data | 8.8 [42] | | Total energy reported for Norway, Sweden, and Denmark but excl. Finland (DNA) |
| Telia | 1135 [36] | 1135 [36] | 21.1 [36] | | |
| Reported total | 14,257 | 12,200 | 331 | | |
| Adjustment | | | 401 | | 70 M subscriptions added based on market statistics |

*3.9. Electricity Consumption and Related Carbon Emissions Estimated for Telecom Network Operators in Europe-30 for 2018*

The electricity consumption of the network operators in Europe-30 can be estimated based on extrapolation of the ETNO data set, as presented in Table 12. To derive this estimate for 2018, the intensities per subscription per network type and including the overhead from Table 7 were used, together with the subscription data from Table 4. Using a number of subscriptions as the basis for extrapolation follows the calculation approach established in Reference [47]. Moreover, to derive the associated carbon emissions, an average EU electricity emission factor of 0.3-kg $CO_2e$/kWh [48] was used for the electricity consumption of all subscriptions but the ETNO ones. This resulted in larger carbon emissions per subscription compared to the ETNO operators, as, taking a conservative approach, no additional purchase of renewable electricity beyond the grid average was considered in the extrapolation. The ETNO Europe's share of the total electricity consumption in 2018 was

37% of the estimated Europe-30 value. Similarly, the ETNO Europe data set corresponds to 22% of the total $CO_2$ emissions estimated for the operators in Europe-30.

**Table 12.** Estimated EU-28 and Europe-30 levels for 2018 based on extrapolation of the ETNO Europe data.

|  | Input Data | Extrapolations | |
| --- | --- | --- | --- |
|  | ETNO Europe 2018 | EU-28 [1] | Total Europe-30 |
| Electricity consumption (TWh) | 14.5 | 38.4 | 39.5 |
| Purchased renewable electricity (TWh) | 10.3 | 10.3 [2] | 10.3 [2] |
| Electricity $CO_2$ emissions (Mt) | 1.8 | 8.7 | 9.0 |
| Other reported $CO_2$ (Mt) [3] | 0.7 | 1.8 | 1.8 |
| Additional estimated $CO_2$ (Mt) [3] | 0.25 | 0.66 | 0.67 |
| Additional fuel and energy supply $CO_2e$ emissions (scope 3) based on LCA (Mt $CO_2e$) | 0.67 | 2.5 | 2.5 |
| Total $CO_2$ emissions (Mt) | 3.36 | 13.5 | 14 |

[1] UK was still part of the EU in 2018. [2] No further purchases of renewables assumed for the extrapolated part. [3] See Figure 8 for the details on 'other reported' vs 'additional estimated'. This includes other site energy, fleet, travel, and commuting.

### 3.10. Electricity Consumption Data for 2020 in Europe-30, including Additional Operators

Additional data for 2020 was collected also from other operators representing 32 national networks within Europe-30, complementing the 2020 data for the ETNO operators introduced in Section 3.8. This data set is outlined in Table 13. Data traffic were only found for Cosmote Romania (2018–2020: 0.17–0.29EB [24]) and for Vodafone's traffic over mobile networks (2018–2020: 3.46–7.48EB [49]).

**Table 13.** Publicly reported values for Europe for the ETNO and other operators for 2020.

| Operator | Electricity (GWh) | Renewables (GWh) | Subscriptions (Million) | Comments |
| --- | --- | --- | --- | --- |
| ETNO Europe operators | 14,257 | 12,200 | 401 | See Table 11 |
| Cosmote Romania | 121 [40] | unclear | 3.6 [41] | Only mobile subscriptions |
| DT excl Germany | 1895 [27] | 1450 [27] | 52.6 [43] | B2B subscriptions not included |
| Orange France [1] | 2186 [50] | 0 [50] | no data | Low power grid electricity |
| Polkomstel | 252 [51] | no data | unclear | |
| Vodafone | 3821 [52] | 1369 [52] | 140 [52] | 2019/2020 data |
| Wind3 | 822 [53] | no data | 20.3 [54] | Only mobile subscriptions |
| Total | 23,354 | (15,019) | 617 | |
| Adjustment | | | 70 | 12 M (Polkomstel), 55 M (Orange) 2.7 M (Wind3, fixed) additional subscriptions estimated based on market statistics |

[1] No data found for Orange Europe outside of France.

Based on Table 13, the total result for the extended 2020 data set was:

- There were 23.4-TWh electricity and 687 million subscriptions resulting in an average electricity consumption per subscription of 34 kWh/subscription. This is lower than the corresponding value for ETNO operators for Europe (36 kWh/subscription). This difference is partly due to a lower share of PSTN and core networks and lower overhead operations (operators' data centers, offices, stores, etc.) among the additional operators.
- In total, 15 TWh of purchased renewable electricity was identified. However, data gaps exist for several operators.
- If scaled by subscriptions to the entire Europe-30, the resulting total electricity consumption is estimated to be about 35 TWh for 2020, covering about 1034 million subscriptions [15–17]. For EU-27, the resulting total electricity consumption is estimated to be about 29 TWh for 2020, covering about 865 million subscriptions [15–17].

## 4. Discussion

### 4.1. General Comments on the Results

The operational electricity consumption changes as the network technology evolves, as implied by Section 3.2. Such evolution processes include inter alia: (i) the shift from fixed to mobile connectivity, (ii) fiber optics replacing copper-based connections, (iii) cloudification of data servers, and iv) the expansion of the Internet of Things and associated Machine-to-Machine connections. Overall, the newer technologies (such as 5G and fiber) are more energy-efficient than older technologies, such as copper-based fixed lines and early generations of mobile networks. Consequently, these provide opportunities for more efficient networks, but the modernization and design of networks still need to consider how to counteract any growth in electricity consumption when their capabilities and capacity demand increase [55,56].

During the period of 2015–2018, the assessed ETNO Operators' overall electricity consumption remained almost flat (just below 1% growth) and so did their total volume of ICT subscriptions (fixed and mobile), as shown in Figure 3. During the same period, the data traffic increased about three times on average based on the data traffic reported by the ETNO Operators (see Section 3.7). This confirms the alignment between electricity consumption and number of subscriptions and the decoupling between electricity consumption and data traffic observed in the authors' previous study of network operators [10]. This indicates that the electricity consumption of networks cannot be derived based on data intensities only. Instead, electricity consumption per subscription seems to be a reasonable basis for extrapolation when estimating the electricity consumption of networks.

The electricity consumption in offices and shops was reduced over the assessed period, as Section 3.2 shows. One reason could be the transition to activity-based and flexible offices that occur in some regions, which reduces the office space per employee and, thereby, the associated energy consumption. In addition, in the past, offices were usually integrated with major technical sites for network operations, data centers, and similar. Today, telecom network operators instead often separate those, giving a clearer boundary between offices and data centers and more energy-efficient office buildings. This development is partly driven by the cloudification of data centers.

The intensities in terms of electricity consumption per subscription presented in Section 3.3 are not directly comparable over time, as the capabilities and performance per user are not constant. Still, these timelines give an indication of how the average energy intensity at the network level developed in 2015–2018, although the driver of any change needs further analysis. For example, the increase in electricity consumption per subscription for mobile networks is associated with the provision of higher data capacity per subscription emerging from newer network generations such as 4G and 5G, while older, less energy-efficient generations such as 2G and 3G have not yet been decommissioned. Typically, new generations are also, in the beginning, serving fewer users than their capacity allows, so initially, their load of independent electricity consumption is shared by fewer subscriptions.

Similarly, differences in the intensities for different types of subscriptions (e.g., fixed and mobile) are not directly comparable and can only be interpreted when considering also their different capabilities. It is also noted that the subscription statistics and trends reported for fixed networks by ITU [15,16] seems to be slightly different and often delayed compared to the actual development, as reflected in the publicly available reports from telecom operators. This is probably due to the ITU reports being derived and aggregated through local national telecommunication authorities and not directly from the telecom operators.

The electricity consumption per subscription for ETNO Europe is higher than for ETNO Total, especially for mobile networks that consume about 50% more electricity per subscription for Europe, as presented in Section 3.3. In contrast, the averages for the fixed networks are just slightly higher than the global averages, mainly due to a low share of fixed networks in the ETNO Overseas data set. In addition, fixed deployments need less adjustments to the environment they are built in, while mobile networks to a larger extent depend on country- and region-specific conditions, such as topology, population, subscriber density, geographies to cover, quality of service requirements, and the share of urban and rural populations. Mobile networks can also be shared in various ways, while fixed networks typically provide unique lines for each customer. However, fixed networks also show some variation in electricity consumption per line due to the age of equipment, type of hardware, and line distance, but the differences are rather seen between different kinds of deployments than between countries.Section 3.3 showed the difference between European and overseas operations for the assessed ETNO Operators with regards to the average electricity consumption per subscription, when averaging over all types of networks, which results in lower values for ETNO Overseas. A main reason for this is the ETNO Operators' focus on mobile network deployments in their overseas operations, which have lower energy consumption per subscription. This is due to mobile networks being more flexible and cost-efficient per subscriber to establish and operate [57].

To derive a longer trend line, Sections 3.6 and 3.7 put the results of this study into the perspective of an older one, with a similar data set provided by the authors for 2010–2015 [10]. One reason for doing so is to explore whether claims regarding an increased use of renewable electricity, increased energy efficiency, and decreased carbon emissions per subscription could be confirmed over a longer timespan. Moreover, technological changes involving changes in the infrastructure (i.e., the telecom networks) are associated with a gradual implementation that often concerns tens of thousands of big and small installations covering vast geographical regions, which makes it interesting to monitor trends over longer periods and to put any short-term developments into the perspective of the longer trend lines.

The publicly available reports from ETNO operators for 2020 show a slight decrease in electricity consumption for the period 2018–2020, while, at the same time, increases the share of renewable electricity, which gives a large reduction in carbon emissions related to electricity usage, as investigated in Section 3.8. This was also confirmed in a report by ETNO where as much as 75.3% of the total energy used was from renewable sources in 2020 [58]. The increased usage of renewable electricity is more prevalent in the data set for European operations. Hence, if a similar electricity supply strategy would be applied by telecom operators globally, it would result in a significant reduction of greenhouse gas emissions for the sector. The difference in prevalence has not been analyzed, but possible reasons include differences in availability, accessibility, policies, and customer demand.

Interpretations of the 2020 values also need to consider the COVID-19 pandemic and the societal lockdowns, which greatly increased the use of digital services during the period. A growth in data traffic of about 95 percent was experienced from 2018 to 2020, as shown in Figure 12 for the ETNO Total. In comparison, the increase in subscriptions was quite moderate at 9 percent, and the overall electricity consumption decreased by 1 percent. This is in line with earlier reported numbers [59]. A reason for this limited increase in electricity consumption may be that older network equipment is being replaced by new equipment with better data capacity and energy efficiency, while, at the same time, network expansion

investments are being put on hold in favor of upgrades of existing networks during these societal lockdowns.

### 4.2. Data Quality

The ETNO data used for this study are considered to represent actual conditions, and the same goes for the data of the previous study [10], which was used to derive the 2010–2018 trends. This view is also shared by the assessed ETNO members. The data quality has increased significantly, and the data gaps are also fewer compared to the earlier study [10]. In the present study, the data for each individual year from 2015 to 2018 have been reported. All operators from the first study participated in the new study, and additional operators were added. In the data set of 2015–2018, about three-quarters of the participating operators reported details regarding their electricity consumption and, accurately, the split and allocation of the electricity consumption between the requested network and activity types. In addition to higher data granularity, the present data set is also considered to be of better quality than the previous study [10] due to the increased energy consumption awareness among the operators and, not least, due to the increased focus on 5G and its energy consumption during recent years [60].

The operators that provided detailed data and allocation between activities and network types are considered as a good enough basis for allocation of the electricity consumption for the remaining operators that only reported the total electricity consumption or a limited set of details. Such allocation was not possible to perform in the previous study [10] due to a too-limited sample that would have resulted in too-uncertain estimates. In total, this study represents a higher total volume of subscriptions, and the data are considered more reliable due to the higher data quality.

### 4.3. Uncertainties

Deriving and allocating the electricity consumption of a network operator with high accuracy is time-consuming and requires deep network knowledge, as well as good and reliable data sources. Even though the study is based on measured data, a detailed allocation, e.g., to fixed and mobile networks, might be difficult to perform for the reporting operators due to the technical and organizational aspects and reporting structures.

Communication networks are geographically distributed, and sites often include equipment for both fixed and mobile operations. Many of these local and rather small sites often use only one electricity subscription and electricity meter. In some cases, such sites might be the dominating type and, by that, also become a significant contributor when deriving and allocating the total electricity consumption for a specific network operator's actual share of fixed or mobile network operations. Hence, allocation between an operator's different network types becomes a source of uncertainty for the overall assessment.

Additional sources of uncertainty are equipment located at sites that are hosted by other operators, aka colocation, and shared buildings where individual electricity subscriptions or meters are missing. In this data set, all operators have reported the measured or estimated electricity consumption of their equipment when hosted by others and subtracted the electricity consumption of other equipment owners' installations that they host themselves. However, this is a manual estimation process, not a measurement, which also brings uncertainties. For renewables, definitions differ between countries, as described in Section 3.1. Due to this difference, an additional uncertainty is that electricity categorized as 'renewable' may include other fossil-free electricity production alternatives in some markets. However, the share that is fossil-free but not renewable is considered limited based on the participating operators and their geographical locations.

### 4.4. Applicability of Results for Other Regions

The reporting ETNO operators' subscriptions represent a substantial share of the Europe-30 subscription base, as detailed in Section 2, but they also represent a significant share of the global subscriptions. Therefore, the reported results and trends might be

considered as a relevant data set for deriving a first approximation of the telecom networks also in other regions.

However, for the most representative data, it is recommended to use operator data from the country or region under study, if available. In our global studies [1,10], we preferred to use data from all larger regions in the world. Still, when this study's results were extrapolated to the global level based on subscriptions, the results were found to be in line with collected but not yet published data for telecom network operators globally in 2020. This indicates that the ETNO data set could be used as proxy data for estimating global conditions if no more specific data is available.

If the present data set is used to model other regions, the following precaution applies. The reported ETNO Operators' data might have better than average energy and climate performance, especially in a global context, as operators with bad performances may be less interested in sharing their data. On the other hand, the ETNO data set includes several incumbent (traditional) operators that typically represent a larger share of a nation's physical networks, especially the core networks, and main operation of older networks like PSTN. Hence, the ETNO data set may be better in terms of renewables but worse in terms of overhead compared to an average operator.

For ICT mature regions like the US, China, Japan, and Korea, the ETNO Europe data is probably a better benchmark than the ETNO Total data, as the higher demand for bandwidth for ETNO Europe results in slightly higher electricity consumption per subscription. Due to the existing higher penetration of mobile subscriptions for ETNO Europe, the network planning needs to include the use of higher frequencies, which demand more electricity due to shorter site-to-site distances. Moreover, a capacity increase for mobile networks costs more from an energy perspective for ETNO Europe than for the overall sample, as discussed in Section 4.1. In parallel, there is a declining electricity consumption trend for fixed network operations due to the continuing process of VoIP replacing PSTN services.

If the results of this study are used as an input for other studies, it is better to use the electricity data and apply local emission factors than to use the presented carbon emission values directly to avoid that a difference in the energy mixes distorts the results.

*4.5. Applicability of Results as a Basis for Estimates of the Future*

This study shows that electricity consumption per subscription has been quite stable over the years for ETNO Europe and overseas operations. Hence, assuming this condition prevails, forecasts of electricity consumption and related carbon emissions can be based on subscription volumes. At the same time, this data set clearly shows the lack of a direct link between the growth in data traffic and electricity consumption. As already discussed in Section 3.8, the electricity consumption decreased in 2020, and it remains to be seen whether this was only due to the pandemic or will have a lasting effect. It is assumed that a slowdown of network expansion, but an increase in replacing older networks, has led to a lower electricity consumption overall. It is also expected that this development will change as society gets back to normal. Note that the changes in electricity consumption so far are small, and it is not known if this observation is limited to ETNO Europe.

Other future developments of interest related to the network operators' electricity consumption and carbon emissions include the ongoing network transformation among many network operators within ETNO Europe where 5G and digital fiber connectivity are deployed while legacy networks such as cupper-based communication (i.e., fixed telephony and copper-based broadband services) and also increasingly older generations of mobile telephony are decommissioned.

*4.6. Alignment with Standards*

The data set developed in this study is intended to be applicable as input when developing the carbon footprint of the ICT networks or of the ICT sector in line with the relevant ITU standards [11,47]. Consequently, this study is not reviewed for full alignment

with these standards, but it is reviewed whether it adheres to the principles that make the data set relevant for studies based on these standards, as outlined in the following sections.

The ETNO data set and the additional data collected respect the principle established by [11] that "the best way to determine the energy consumption of ICT goods during the use stage is, whenever possible, to measure a large number of ICT goods operating in a live network or products in real live operating environments over a long period of time (e.g., a year to capture all aspects of variations in traffic, temperatures, different use behavior, climate etc.)".

Moreover, in line with Reference [47], the paper provides details on the time horizon, geographical coverage, system boundaries (the network part of the ICT sector), and provision of absolute carbon emissions. The paper has not defined a reference flow or functional unit, but it would be possible to establish one if of interest to subsequent studies in line with "the total operation of ETNO Operators within and outside Europe during 4 consecutive years excluding other life cycle stages". The study also collects recommended contextual data and establishes one recommended complementary reference flow (per subscription) and explores another (per data). The principles regarding timeliness, accuracy, and accessibility are respected. It is also clearly defined that this data set provides input on the use stage only. The study also refers to actual emission factors of operators and explicitly states the emission factor used for the extrapolations. However, for emission factors, the electricity supply chain and distribution losses account separately instead of being integrated in carbon emission values for electricity, energy, and fuels, as demanded by the standard. Instead, it follows the principles for company reporting established by the GHG Protocol, where those are accounted for separately [13]. Thus, if this data set is reused for studies according to References [11,47], the emission factors need to be revisited.

On a more detailed level, the data set includes primary data for both grid electricity and self-generation. It also takes note of the principles for site sharing. As far as green certificates are concerned, the study considers actual certificates but makes a conservative model without certificates for the extrapolated part in line with the standard. The paper also sees subscriptions as a proxy for the user, as established in the standard. Finally, the paper also tries to make a careful analysis of the trends rather than just presenting them.

### 4.7. Future Work and the Importance of Collecting Up-To-Date Data from Real Networks

Telecom networks are complex and develops rapidly, which emphasizes the importance of regularly collecting measured, up-to-date data from real networks. Transparently communicating such data and the used methodology gives others the possibility to compare and base their ICT calculations and forecasts on actual conditions rather than on estimates based on product data sheets and theoretical models. An assessment of the actual behavior of telecom networks and their components is important to understand the priorities and to drive operators' decarbonization strategies but, also, to create a reasonable understanding of the footprint of telecom networks in relation to other carbon emission hotspots. Understanding historical trends and the connection between different parameters such as electricity consumption, carbon emissions, data traffic, and subscriptions at different levels would also provide a relevant basis for predicting future developments overall, as well as for different network types, which would enable more informed business and policy decisions.

This study focuses on operational emissions, but from a life cycle perspective, there are additional upstream embodied emissions covering the raw material acquisition, production, and transportation prior to the use, as well as downstream emissions related to end of life. For a complete picture of the total carbon emissions of networks, the embodied emissions need to be studied as well (These are expected to be in line with an additional 16% for world average electricity, as outlined in Section 2). Moreover, measured data should also be collected from additional network operators and other regions. With the increasing focus globally on mitigating climate change and on reporting climate data, operators are likely to increase the granularity of their public reporting, which will make this kind of studies

easier to perform. Although data measured over long time periods (e.g., per annum) give the best knowledge about the long-term performance of the assessed operators, studies based on data with higher spatial and temporal granularity would be a complement that could enable a statistical analysis of the data sets, thereby allowing for a deeper analysis of the results with regards to variations due to changing conditions for and between operators.

## 5. Conclusions

This paper presents details regarding the electricity consumption and carbon emissions of telecom network operators with headquarters in Europe for the period of 2015–2018. The operators represent 8.2% of global subscriptions and 36% of European subscriptions. The operators are identified, but the data set is presented at an aggregated level for their operations in Europe and overseas. The aggregated data for Europe is compared to publicly available but less granular values for the assessed operators, which shows that the assessed data set is well-aligned with the overall data.

The key findings of the study are:

- The electricity consumption and number of subscriptions for the reporting telecom network operators (ETNO Total) remained nearly constant (+1 percent and -3 percent, respectively) between 2015 and 2018, while the data traffic increased by a factor 3.
- During the period, the share of renewable electricity increased to 71%, which has led to a decrease in carbon emissions. For 2018, the operational carbon intensity of the operators equaled 6.6-kg $CO_2$ eq per subscription (excluding embodied emissions).
- The intensities were quite constant for 2015–2018. For 2018, the electricity intensity per subscription (including overhead) was 62 kWh/subscription for fixed networks and 17 kWh/subscription for mobile networks, leading to a total average of 26 kWh/subscription.
- For the extended period of 2010–2018, the electricity consumption per subscription (ETNO Total, including overhead) remained quite stable and slightly below 30 kWh/subscription despite substantial data traffic growth (by a factor 12).
- For Europe, the intensity for the reporting telecom network operators (ETNO Europe) was 39 kWh/subscription for 2018, representing a slight increase over the period. Based on public reporting by ETNO Operators for 2020, the intensity then reduced slightly (36 kWh/subscription), which could be compared to 34 kWh/subscription for the extended data set covering ETNO Europe and additional operators.
- The overall electricity usage and operational carbon emissions of telecom operators are estimated as 38 TWh and 14 Mton $CO_2$ eq for the EU (then including the UK) for 2018, and for 2020, the electricity consumption is estimated as 29 TWh for EU-27 (excluding the UK).

The summarized results of the aggregated collected data from the ETNO Operators can be found in the Supplementary material.

**Supplementary Materials:** The following supporting information can be downloaded at: https://www.mdpi.com/article/10.3390/su14052637/s1, Table S1: Summarized results of aggregated collected data from ETNO Operators Questionnaire version 14 May 2019.

**Author Contributions:** Conceptualization, D.L., J.M., N.L., and P.B.; methodology, D.L. and J.M.; validation, N.L. and P.B.; formal analysis, D.L. and J.M.; data curation, D.L. and J.M.; writing—original draft preparation, D.L. and J.M.; writing—review and editing, D.L., J.M., N.L., and P.B.; and visualization, J.M. All authors have read and agreed to the published version of the manuscript.

**Funding:** This research received no external funding.

**Institutional Review Board Statement:** Not applicable.

**Informed Consent Statement:** Not applicable.

**Data Availability Statement:** Not applicable.

**Acknowledgments:** The authors wish to thank all the named telecom network operators that generously contributed with the data and measurements and, by that, made this study possible. We would also like to thank ETNO for their valuable support in this data collection and our colleagues at the KTH Royal Institute of Technology in Stockholm, as well as our colleagues at Telia Company and Ericsson, for the continuous support and interest in the research and findings.

**Conflicts of Interest:** The authors declare no conflict of interest.

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
