# Peer review of "Electricity Consumption and Operational Carbon Emissions of European Telecom Network Operators"

_sustainability, doi:10.3390/su14052637_

Round 1
Reviewer 1 Report
Sustainability reviewer’s comments
Dear authors
This paper presents electricity consumption and COâ‚‚ emissions of the part of ICT sector in Europe and other area from ETNO data.
It is very significant information for ICT sector to consider the sustainability management in the world. This paper used many useful data and provided important results.
However, there are some points which are needed to confirm and correct before publication.
Those are:
- This paper has too many keywords. Please confirm the rule of the keywords. (Keywords: Three to ten pertinent keywords need to be added after the abstract. We recommend that the keywords are specific to the article, yet reasonably common within the subject discipline.) (https://www.mdpi.com/journal/sustainability/instructions)
- In “Introduction” section gave the many information of previous studies and reports. However, I couldn’t find the main purpose of this research. Please write the motivation with the novelty of this paper.
- Figure 4 (a), Average/residual rid mix occupy 82% in ETNO Overseas. If possible, please explain which country or the area caused by the results and add the information.
- Line 498, the number 3 is capital letter. Please revise the small letter.
- In “Results” section, totally the results in Europe are better than other area. Therefore, the other area should take action to solve the problem. If possible, please provide the suggestion.
- The technology of the telecom network operators and subscription will be improved for energy efficiency, data traffic and so on. I consider the demand of the electricity will rapidly increase in the future. If possible, please consider the any suggestion or solution for this problem with the information from this paper.
I would like to confirm above the review and revise the manuscript.
Best regards,
Author Response
Dear Ms Wang and and reviewers
Please see the attachment in which we have included both reviewers comments labeled as R1 (Reviewer 1) and R2 (Reviewer 2).
An updated article version including "Track changes" relevant for the reviewers comments are attached. However if there's a need for the updated article in full as well, including "track changes" for all updates compared to the first version, please just let us know and we will upload that version as well.
Yours sincerely//Dag

Reviewer 2 Report
This paper presents measured and aggregated operational electricity consumption and greenhouse gas emissions for named European telecom network operators for the period 2015 – 2018. Moreover, these results are put in relation to the author’s previous research covering the period 2010-2015. The present study is built on an extensive primary data set, collected from members of European Telecommunication Network Operators (ETNO) covering their operations in Europe and beyond, complementing publicly available reporting by providing data with higher granularity. The collected data set corresponds to roughly 36 percent of European subscriptions and about 8 percent of global subscriptions. The results include total and allocated electricity consumption, related carbon emissions, other emissions of the assessed operators and corresponding intensities for 2015-2018, as well as derived estimates for Europe and for 2020. The study concludes that the electricity consumption and number of subscriptions for the reporting telecom network operators remained nearly constant (+1 percent and -3 percent respectively) between 2015 and 2018, while data traffic increased by a factor 3. For the extended period 2010-2018 the electricity consumption per subscription remained quite stable, slightly below 30 kWh/ subscription despite substantial data traffic growth (by a factor 12).
This paper is with some merits for Sustainability, however, it requires major revisions.
Firstly, the abstract should be refined to clearly indicate what authors had done within 150 words.
Secondly, for Section 1, authors should provide the comments of the cited papers after introducing each relevant work. What readers require is, by convinced literature review, to understand the clear thinking/consideration why the proposed approach can reach more convinced results. This is the very contribution from authors. In addition, authors also should provide more sufficient critical literature review to indicate the drawbacks of existed approaches, then, well define the main stream of research direction, how did those previous studies perform? Employ which methodologies? Which problem still requires to be solved? Why is the proposed approach suitable to be used to solve the critical problem? We need more convinced literature reviews to indicate clearly the state-of-the-art development.
For Section 2, authors should introduce their proposed research framework more effective, i.e., some essential brief explanation vis-à-vis the text with a total research flowchart or framework diagram for each proposed algorithm to indicate how these employed models are working to receive the experimental results. It is difficult to understand how the proposed approaches are working.
For Sections 3 and 4, authors should use more alternative models as the benchmarking models, authors should also conduct some statistical test to ensure the superiority of the proposed approach, i.e., how could authors ensure that their results are superior to others? Meanwhile, authors also have to provide some insight discussion of the results. Authors can refer the following references for conducting statistical test.
Forecasting short-term electricity load using hybrid support vector regression with grey catastrophe and random forest modeling. Utilities Policy, 2021, 73, 101294.
Author Response

(The authors gave the same response as above.)

Round 2
Reviewer 2 Report
Authors have completely addressed all my concerns.